# On-chip silicon electro-optical modulator with ultra-high extinction ratio for fiber-optic distributed acoustic sensing

Zhuo Cheng[1,6], Xiaoqian Shu [1,6], Lingmei Ma [1,6], Bigeng Chen [1] ✉, Caiyun Li [1], Chunlei Sun[2], Maoliang Wei[3], Shaoliang Yu [4], Lan Li [2], Hongtao Lin  & Yunjiang Rao [1,5] ✉

Ultra-high extinction ratio (ER) optical modulation is crucial for achieving high-performance fiber-optic distributed acoustic sensing (DAS) for various applications. Bulky acousto-optical modulators (AOM) as one of the key devices in DAS have been used for many years, but their relatively large volume and high power consumption are becoming the bottlenecks to hinder the development of ultra-compact and energy-efficient DAS systems that are highly demanded in practice. Here, an on-chip silicon electro-optical modulator (EOM) based on multiple coupled microrings is demonstrated with ultra-high ER of up to 68 dB while the device size and power consumption are only $260 \times 185\ \mu m^2$ and 3.6 mW, respectively, which are at least two orders of magnitude lower than those of a typical AOM. Such an on-chip EOM is successfully applied to DAS with an ultra-high sensitivity of $-71.2$ dB $rad^2$/Hz (4 pε/√Hz) and a low spatial crosstalk noise of $-68.1$ dB $rad^2$/Hz, which are very similar to those using an AOM. This work may pave the way for realization of next-generation ultra-compact DAS systems by integration of on-chip optoelectronic devices and modules with the capability of mass-production.

Fiber-optic distributed acoustic sensing (DAS) based on phase-sensitive optical time domain reflectometry (Φ-OTDR) has enabled a wide range of practical applications such as infrastructure monitoring[1,2], peripheral security[3,4], and seismicity[5,6]. Bedsides, the DAS technology also provides an exceptional approach for multiple natural scientific research directions[7] ranging from insect monitoring[8], ice activity study in Arctic sea[9] to optical microphone[10], etc. Such scientific DAS applications indicate the exceptional capability of observing the nature via acoustic waves. In Φ-OTDR, pulsed probing light is launched into sensing fiber to locate disturbance events along fiber length[11]. Usually, the optical pulses are generated by externally modulating a narrow-linewidth continuous wave (CW) laser to insure a high coherency for precise phase demodulation of Rayleigh back-scattering (RBS) signal in the sensing fiber. Beside the modulated optical pulses, the leakage CW light induced by finite extinction ratio (ER) of the modulation also interacts with the sensing fiber[12]. The corresponding back-scattering light is continuous in time domain and interferes with the signal from the pulses, which influences detection limit as well as signal-to-noise ratio (SNR) of sensing[13]. Therefore, optical modulators with ultra-high ER (>55 dB) are of great importance for realization of high-performance DAS systems.

[1]Research Center for Optical Fiber Sensing, Zhejiang Laboratory, Hangzhou 311100, China. [2]Key Laboratory of 3D Micro/Nano Fabrication and Characterization of Zhejiang Province, School of Engineering, Westlake University, Hangzhou 310024, China. [3]College of Information Science and Electronic Engineering, Zhejiang University, Hangzhou 310027, China. [4]Research Center for Intelligent Optoelectronic Computing, Zhejiang Laboratory, Hangzhou 311100, China. [5]Key Laboratory of Optical Fiber Sensing and Communications (Education Ministry of China), University of Electronic Science and Technology of China, Chengdu 611731, China. [6]These authors contributed equally: Zhuo Cheng, Xiaoqian Shu, Lingmei Ma. ✉e-mail: chenbg@zhejianglab.com; yjrao@uestc.edu.cn

Currently, acousto-optical modulators (AOMs) are widely applied in commercial DAS systems[14,15] due to their high ER. However, the bulky nature (size in centimeters) and large power consumption (≥1 W) are becoming the bottlenecks of further improvement of system compactness and power efficiency, which are critical for DAS applications where device size and power consumption are essential, such as on small platforms. Furthermore, the cost of AOMs needs to be reduced significantly in order to meet the mass application requirement of DAS industry. Hence, on-chip integrated optical modulators can be considered as next-generation ultra-compact modulators to overcome the drawbacks of AOMs mentioned-above and meet higher requirements in terms of device size and power consumption. Chip-scale AOMs are being intensively studied to explore acoustic-optic interactions in highly confined volumes[16,17], but not yet ready for application to DAS. On the other hand, electro-optical modulators (EOMs) based on silicon photonics featuring high compactness, high speed, and low energy cost[18–20] (~2 pJ/bit, i.e., 0.1 W for 50 Gbit/s) have been widely developed and accepted for commercial transceiver modules and optical communication systems[21]. However, it is also noticed that the ER presented in dynamic modulation (especially at high speed) of those EOMs are usually less than 10 dB although static characterization can show much higher values[22,23]. Realization of ultra-high ER optical modulation based on on-chip EOMs is challenging and reports of their applications for DAS are still in absence.

In this paper, we demonstrate on-chip silicon EOMs with ultra-high ER up to 68 dB in experiment. The EOMs consist of four serially-coupled microrings fabricated on silicon-on-insulator (SOI) wafers. The couplings between the microrings and add/drop waveguides are designed delicately to form a bandpass filter with a high out-of-band rejection ratio. Carrier injection scheme is employed for electro-optical modulation by embedding PIN (i.e., positively-doped, intrinsic, and negatively-doped) structures[24] as phase shifters in the microrings. In this way, transmittance modulation of the EOM with substantial contrast is achieved by shifting the passbands with the PIN phase shifters. Ultra-high ER of 68 dB from the modulated optical pulses is obtained with a self-heterodyne measurement. The EOM size on chip is only ~260 × 185 μm² including the electrodes, in contrast to the centimeter-scale acousto-optic crystal in an AOM. Benefitting from the steep passband roll-off and efficient PIN phase shifters, driving voltage for ultra-high ER modulation is less than 1 V. The corresponding power consumption is evaluated to be 3.60 mW which is two orders of magnitude lower than that of an AOM. The measured rise and fall time of the modulated pulses are 6.54 ns and 0.40 ns, respectively. Such responses are fast enough for typical DAS systems in which pulse duration is usually about 10–100 ns. The EOM is then packaged and successfully applied in a DAS system. For 2-km single-mode sensing fiber, the measured sensitivity is up to 4 pε/√Hz (−71.2 dB rad²/Hz) which is the same as that of a system in an identical configuration except for the usage of a state-of-the-art commercial AOM (>65 dB ER). The spatial crosstalk levels are also very close for both systems. Thanks to the adjustable ER of the EOM, the sensitivity and crosstalk dependences on ER are directly obtained by experiment. A 28-dB suppression on spatial crosstalk noise (SCN) is achieved with 40-dB ER enhancement, clearly revealing the contribution of an enhanced ER to DAS performances. The demonstrated on-chip silicon EOM is highly promising to facilitate the development of ultra-compact, low-power, and low-cost DAS systems.

## Results

### Device design, fabrication, and characterization

The proposed coupled microring EOM includes four 10-μm-radius rings as well as the add/drop waveguides on 220-nm SOI, as illustrated in Fig. 1a. The rib waveguide width is 450 nm for fundamental TE mode operation while the rib height is 130 nm with 90-nm-thick slab for doping and electrical conduction. To realize a steep roll-off and flat-top bandpass filter, the coupling efficiencies between the rings and add/drop waveguides are set as 0.269, 0.01, 0.004, 0.01, and 0.269 from left to right, respectively, according to the synthesis principle of a high-order optical bandpass filter (ref. 25, also see Supplementary Note 1). Then, the corresponding physical gap sizes (151, 493, 587, 493, and 151 nm) between the coupling waveguides are determined by FDTD simulations. PIN structures are employed in parts of the rings, which are formed by heavily doped P and N regions in the slab and the intrinsic rib waveguides in the middle (Fig. 1b). The doped regions are 600 nm away from the rib edges to minimize optical loss. Ground-signal-ground (GSG) electrode configuration is adopted to arrange the metallic contact pads. The Signal pad in the middle is connected to the P region inside the rings while the Ground

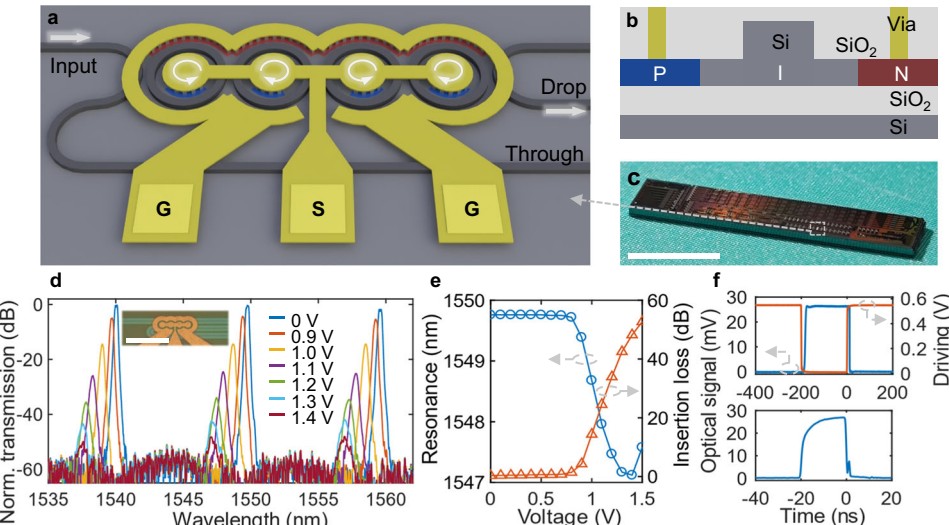

**Fig. 1 | Design and static characterizations of the electro-optical modulator (EOM). a** Schematic illustration of the proposed coupled microring EOM. **b** Cross-section of the PIN structures. **c** Photograph of a delivered photonic chip including the coupled microring EOMs from the multi-project wafer (MPW). Scale bar: 5 mm. **d** Transmission of the fabricated EOM with different DC voltages. Inset: Micrograph of an on-chip EOM. Scale bar: 50 μm. **e** Passband position near 1550 nm and insertion loss dependent on DC voltage. **f** Upper panel: Modulated optical pulse signal with 200-ns duration from the EOM and the corresponding electrical driving signal. Lower panel: 20-ns optical pulse signal from the EOM.

pads are for the N region outside. The EOM size on chip is ~260 × 185 μm², most of which is occupied by the electrodes. In contrast, acousto-optic crystals in AOMs is usually in centimeter-scale. The EOM chips are fabricated via multi-project wafer (MPW) process based on 150-nm CMOS technology. The process flow includes DUV lithography, dry etching, ion implantation, metal deposition, etc., which are fully compatible to large-scale manufacturing. The photograph of one delivered chip including the EOMs is shown in Fig. 1c.

The static transmission spectrum of the fabricated EOM is characterized using a tunable laser (Santec TSL-710) with a synchronized optical power meter (Santec MPM-210). DC voltage is applied to the electrodes to inspect the spectrum shifting. As shown in Fig. 1d, the transmission measured from the drop port shows a bandpass filtering feature at zero bias, presenting a steep roll-off about 100 dB/nm (3 dB to 30 dB decay[26]) and a large out-of-band rejection over 60 dB. Near 1550 nm, the 3-dB bandwidth of the passband is 0.25 nm while the free spectrum range (FSR) is 9.81 nm. The on-chip insertion loss is less than 0.5 dB after normalization to reference waveguides. As the forward bias (output from a source meter, Rigol DP832) increases from 0 V to 1.4 V, the transmission spectrum gradually blue shifts accompanied with significant attenuation. The PIN phase shifters in the rings have facilitated carrier injection through the rib waveguides perpendicularly under forward bias[27]. Hence, the induced substantial growth of carrier density leads to decreased refractive index and increased insertion loss (Fig. 1e) for the waveguides[28]. Near 1 V, the resonance shift efficiency slope is about 5.6 nm/V, corresponding to a $V_\pi$ of 0.9 V. Considering that the phase shifter length in each ring is about 35.3 μm in average, the derived modulation efficiency $V_\pi L$ is about 0.0032 V cm. Beyond 1.4 V, the passband shifting turns inversely to red shift, which is owing to that the stronger thermo-optic effect induced by the electrical heating has outcompeted the free carrier plasma dispersion effect[29]. The measured DC response of the EOM suggests that high-ER modulation can be achieved with a small driving voltage <2 V.

To demonstrate optical pulse generation using the on-chip EOM, we apply a pulsed driving signal to the device with a CW laser input at 1549.82 nm wavelength and detect the output optical signal. The electrical driving has a large duty cycle at high level (red curve in upper panel of Fig. 1f). Hence, the EOM remains in a lossy state except for the generation of optical pulses when the transmission is resumed for a certain amount of time. For example, a driving with 99.8% duty cycle and 10 kHz repetition yields 200-ns pulse duration. The driving voltage set in an arbitrary wave generator (AWG, Siglent SDG6052X) is 0.57 V. Then, the waveform of the pulse is detected by a high-speed photodetector (Thorlabs DET08CFC) and observed in an oscilloscope (Tektronix MSO64B) as given in upper panel of Fig. 1f (blue curve). The rise time and fall time are 6.54 ns and 0.40 ns, respectively, according to the 10−90% edge measurement results of the oscilloscope (6-GHz bandwidth). Because of the inverted driving scheme, the onset of the optical pulse corresponds to the moment when the electrical level returns to the low level of 0 V. As the intrinsic carrier relaxation time is only about 1 ns in silicon waveguides[30,31], the extra time for the rising edge may be induced by the incomplete recovery of thermo-optic effect during the inverted modulation. As demonstrated in lower panel of Fig. 1f, shorter optical pulses (20 ns) can also be realized by adjusting the driving duty cycle. A small spike is observed next to the falling edge, which is owing to a little instability of the driving signal with sharp edges. For duration less than 10 ns, the pulse peak intensity will be influenced. Electro-optical bandwidth of the EOM is characterized with a vector network analyzer (Keysight E5080B), which turns out to be about 0.72 GHz (Supplementary Note 2). Note that the measured bandwidth can not directly correspond to the pulse response using the RC model as they are obtained in different conditions.

In order to verify the capability of ultra-high ER modulation, we firstly perform a static EOM transmission measurement at a single wavelength (1549.74 nm) with increasing DC bias. As shown in Fig. 2a, the transmission starts decreasing rapidly when the voltage exceeds

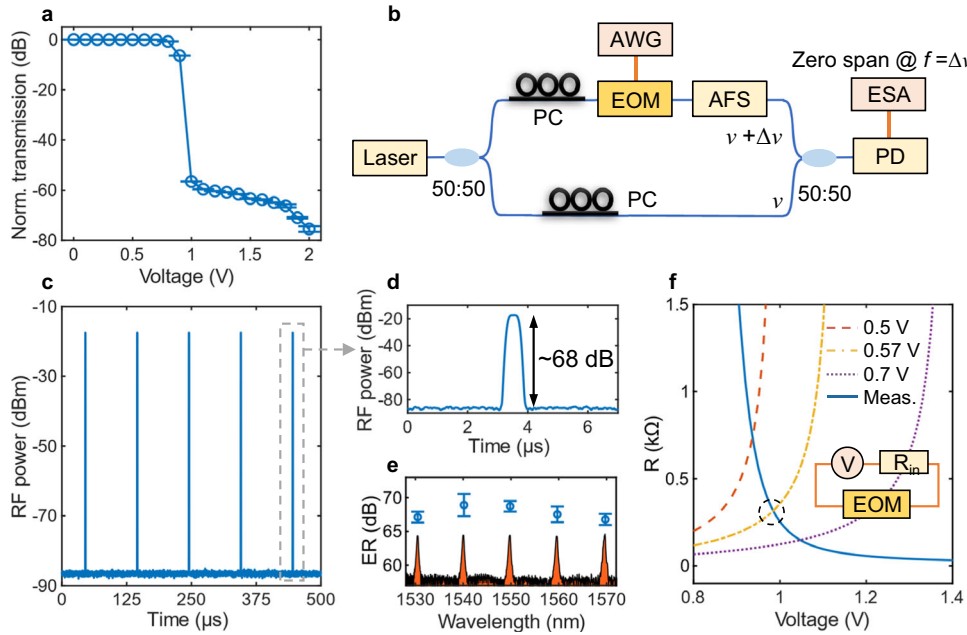

**Fig. 2 | Ultra-high extinction ratio (ER) measurement. a** Static transmission of the electro-optical modulator (EOM) at 1549.74 nm wavelength dependent on DC bias. The lengths of the error bars indicate twice of standard deviations of the results obtained from repeated measurements. **b** Schematic of the self-heterodyne measurement. PC: polarization controller, AWG: arbitrary wave generator, AFS: acousto-optical frequency shifter, ESA: electrical spectrum analyzer, PD: photodetector. **c** Modulated optical pulse train with an ultra-high ER of 68 dB. **d** Waveform of a zoom-in pulse observed in the ESA. **e** Measured ERs for five passbands with corresponding channel spectrum (not to scale). The lengths of the error bars indicate twice of standard deviations of the results obtained from repeated measurements. **f** Calculated $R_{EOM}$ for different $V_{AWG}$ (0.5 V, 0.57 V, and 0.7 V) and measured $R_{EOM}$ dependent on voltage (solid line). The black dash circle marks the intersection between the two $R_{EOM}$ curves according to $V_{AWG} = 0.57$ V in experiment. Inset: A simple circuit model of the driven EOM in serial connection with $R_{in}$.

0.8 V. The output contrast reaches up to 71 dB at 1.9 V, indicating a concrete basis for ultra-high ER modulation. However, it is difficult to directly observe an ER over 30 dB from a signal waveform displayed in an oscilloscope due to the limit of dynamic range. Therefore, a self-heterodyne measurement is set up to show the modulated optical pulses with ultra-high ER[32]. The schematic of the setup is given in Fig. 2b. The optical pulses from the on-chip EOM pass through an acousto-optical frequency shifter (AFS) and interfere with a CW reference light that is from the same laser source. The beat signal of the interference is then collected by a photodetector and received by an electrical spectrum analyzer (ESA). As the beat frequency is fixed by the AFS, one can obtain a temporal response that is basically proportional to the optical pulse waveform by using the ESA (Rohde & Schwarz FSW26) in zero-span mode at the beat frequency (see Supplementary Note 3 for more detail). Thanks to the large dynamic range (>100 dB) of the ESA, we are able to measure the actual ER from the observed pulse waveform. Figure 2c shows the obtained optical pulse train with an ultra-high ER of 68 dB (detected RF power from −85.5 dBm to −17.4 dBm). To the best of our knowledge, this is the highest ER achieved by an on-chip optical modulator, which is measured from dynamic response instead of static transmission contrast. The zoom-in waveform is given in Fig. 2d. Due to the limited resolution bandwidth (10 MHz) of the ESA, the pulse edges displayed in the ESA are not as steep as those in the oscilloscope (Fig. 1f). Featuring multiple passbands in the transmission (Fig. 1d), the EOM supports ultra-high ER modulation at different wavelength channels simultaneously. As shown in Fig. 2e, all the ERs for the five adjacent channels around 1550 nm exceed 65 dB. The corresponding channel spectrum is also given schematically below.

While conventional measure of EOM energy cost is joule per bit referring to optical communication[18,19], we use power consumption to evaluate the efficiency of the ultra-high ER silicon EOM, which is more comparable with that of an AOM. As mentioned above, large duty-cycle (>99%) driving is applied to our EOM to produce small duty-cycle (<1%) pulses for DAS. Therefore, the driving can be regarded as a DC output and thus the power is approximately the product of output voltage and current. However, the resistance of the EOM changes dramatically with voltage (blue solid lines in Fig. 2f) according to an I-V measurement. At the meantime, the AWG output is connected with a 50-Ω internal resistor $R_{in}$ (inset of Fig. 2f), which means twice of the set voltage is applied to $R_{in}$ and external load in serial connection. Then, we use a numerical approach to determine the actual voltage applied to the EOM in serial connection with $R_{in}$ during modulation. According to Ohm's law, the relation $R_{EOM} = V_{act} \times R_{in}/(2V_{AWG}-V_{act})$ can be deduced, where $V_{act}$ is the actual voltage on the EOM, $R_{EOM}$ is the EOM resistance at $V_{act}$, $V_{AWG}$ is the voltage set in the AWG. The calculated $R_{EOM}$ dependence on $V_{act}$ for different $V_{AWG}$ (dash lines in Fig. 2f) are plotted together with the measured $R_{EOM}$. In experiment, the driving voltage set in the AWG is 0.57 V for ultra-high ER pulse generation. According to the intersection of calculated $R_{EOM}$-$V$ (orange dash line) and measured $R_{EOM}$-$V$ (blue solid line) in Fig. 2f, $V_{act}$ is found to be about 0.99 V, corresponding to about 3.65-mA current and 3.60-mW power. The power (0.67 mW) consumed by the internal resistor only adds up a little to the total AWG output power. Such a power cost of the EOM is two orders of magnitude lower than that of a typical AOM (≥1 W), which is mainly attributed to the steep roll-off of the bandpass filtering feature and the efficient carrier-injection modulation with the PIN structures.

Furthermore, we investigate the dynamic response of the EOM in the wavelength range of spectrum shifting. The self-heterodyne measurement is performed when a square pulse signal with 50% duty cycle is applied to the EOM. Then the temporal waveform is recorded for each wavelength (10-pm step) and plotted as shown in Fig. 3a. The driving signal is also depicted at the bottom (white solid line). Beside the pronounced blue shift of the passband at voltage-on, slight red and

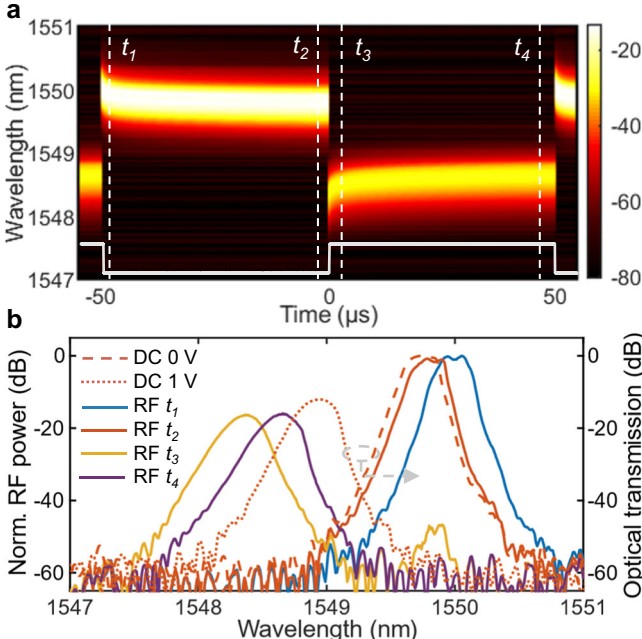

**Fig. 3 | Thermo-optic effect during the electro-optical modulation. a** Dynamic response of the electro-optical modulator (EOM) as functions of wavelength. **b** Dynamic spectra at the moments of $t_1$ to $t_4$ with static optical spectra at 0-V and 1-V bias.

blue shifting with time are also observed at the rising and falling of voltage, respectively. The passband position tends to stabilize until the next voltage switching. This phenomenon can be well explained by the device heating owing to the electrical driving[29]. When the driving level is lifted up to 1 V (from $t_2$ to $t_3$), the passband blue shifts from 1549.78 nm to around 1548.37 nm immediately due to the carrier injection. As the thermo-optic effect in silicon accumulates with time, the passband red shifts a little and then slowly stabilizes (from $t_3$ to $t_4$). Similarly, an inversed electro-thermo-optics process (from $t_1$ to $t_2$) occurs when the driving level is switched back to 0 V, which leads to a small blue shift. The cutline profiles of $t_1$ to $t_4$ given in Fig.3b clearly present the differences between the dynamic spectra at different moments. The spectra of the EOM under DC driving (0 V and 1 V) are also given for direct comparison. The misalignment between the static and dynamic ($t_2$ and $t_4$) spectra suggests longer thermal equilibrium time for both cool and heated states than the pulse duration of 50 µs. It also explains the lower driving voltage (0.99 V) required for narrow optical pulse generation than that for the static characterization (>1.50 V in Fig. 2a) for a similar ER. Under DC driving, electro-optic effect and thermo-optic effect in the EOM compete with each other, leading to a slower spectrum shift at higher voltage (Fig. 1d). As for the pulse modulation, driven by a large duty-cycle signal (>99%), the EOM is under a thermal background and the pulse duration is too short (<1 µs) for a complete thermo-optic response. Therefore, the dynamic modulation is more efficient than changing device transmission by DC driving. For the same reason of thermal background, the operation wavelength is also slightly adjusted to optimized the ER.

## Device application in DAS system

To test our EOM for DAS application, a Φ-OTDR based on imbalanced Michelson interferometry[33] is set up as illustrated in Fig. 4a. An on-chip device with the same structural parameters as described above is fixed into a 3 cm × 2 cm × 1 cm package (inset of Fig. 4a) and then used to generate 100-ns-duration optical pulses as probe light. A thermo-electric plate and a thermistor are also put inside the package for temperature control. The single-mode sensing fiber is 2 km long, in the

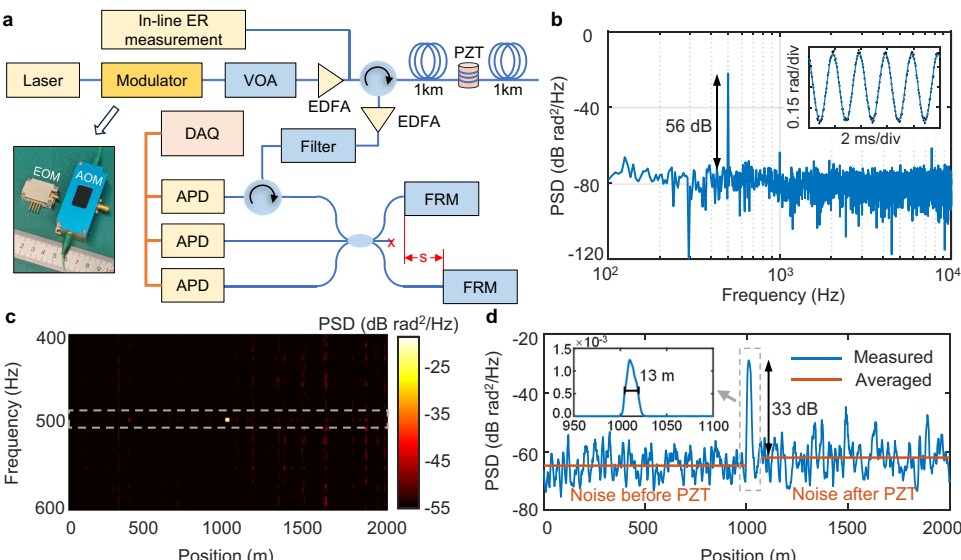

**Fig. 4 | Distributed acoustic sensing (DAS) with the equipped on-chip electro-optical modulator (EOM). a** Schematic diagram of a phase-sensitive optical time domain reflectometry (Φ-OTDR) setup for DAS application. The symbol "s" denotes 5-m fiber length difference for 10-m roundtrip delay. VOA: variable optical attenuator, EDFA: erbium-doped fiber amplifier, PZT: piezoelectric transducer, FRM: Faraday rotation mirror, APD: avalanche photodetector, DAQ: data acquisition. Inset: Photograph of the packaged EOM and a commercial acousto-optical modulator (AOM). **b** Power spectrum density (PSD) of demodulated phase change signal. Inset: corresponding temporal profile. **c** PSD along the full length of sensing fiber from 400 Hz to 600 Hz. **d** Averaged PSD near driving frequency along the fiber. Inset: Zoom-in PSD in linear scale around the PZT position.

middle of which a piezoelectric transducer (PZT) is wrapped by a section of fiber (15 m). Interference of the self-delayed RBS signal after circulation and amplification is implemented by a 3 × 3 coupler. The fiber before one of the Faraday rotation mirrors (FRM) acts as the delay line, which is 5 m here for 10-m roundtrip delay. Three avalanche photodetectors (APD) are used to detect the interfered signal from the coupler outputs. Acousto-optic interaction is produced by generating mechanical vibration on the wrapping fiber with the electrically-driven PZT. The produced strain changes the fiber refractive index and hence the optical phase of light passing through. Then, the phase change and corresponding strain can be obtained from the RBS signal by using the common I-Q demodulation algorithm[34].

As shown in Fig. 4b, using the on-chip EOM in the DAS system, we successfully obtain the dynamic phase signal (black dots) that well coincides with the 500-Hz sinusoidal waveform (blue line) of the driving applied to the PZT. The power spectrum density (PSD) at the PZT position shows an SNR of 56 dB when the phase change amplitude is 0.5 rad corresponding to a strain change of 7.5 nε. Furthermore, PSD is calculated all along the 2-km fiber to inspect the spatial crosstalk as given in Fig. 4c. We can see the striking signal around the PZT position at 1 km as well as some weak crosstalk, which is owing to random interference fading of RBS in the fiber[35]. The PSD results from 490 Hz to 510 Hz (outlined by the grey dash frame in Fig. 4c) are averaged and plotted against position along the fiber, as shown in Fig. 4d. Compared with the signal intensity of −29 dB rad²/Hz, the averaged noise level before the PZT is about −65 dB rad²/Hz, making the averaged spatial crosstalk as low as −36 dB. After the PZT, the noise level slightly increases because of non-negligible connection losses between the stretching fiber and the two fiber spools. The full width at half maximum of the signal in linear scale is about 13 m (inset of Fig. 4d) that generally agrees with the fiber length on the PZT. Using the system with the on-chip EOM, the impact of ER on DAS performances is studied carefully. The ER of modulated optical pulses can be flexibly controlled by changing the suppression on the background CW light with driving voltage. We characterize the ER in line by coupling a small portion of the amplified pulses to the self-heterodyne measurement (Fig. 4a). Then, the system is used to detect the same acoustic signal

under different ERs. The measurement for each ER is repeated for 10 times. In each measurement, PSD values at 1100 evenly-distributed positions (0.8-m step) along the fiber are taken before or after the PZT as SCN. Statistical distributions of all the recorded SCN are presented in histograms as shown in Fig. 5a, where three results with the EOM and one with an AOM are given. The distributions are fitted by Gaussian function[36] and the PSD values for the maximum probability densities on the fitted curves are taken as measures of the system crosstalk performance. One can see that the SCN distribution moves downward obviously as the ER increases. From the histograms, the SCN before (blue) and after (orange) the PZT are also clearly distinguished. As a result, the SCN obtained with the EOM at the highest ER is down to −68.1 rad²/Hz before PZT (−63.7 rad²/Hz after PZT), which is almost the same as that with the AOM (−69.9 rad²/Hz before and −66.7 rad²/Hz after). We notice that the ERs are slightly improved after amplification by the erbium-doped fiber amplifier (EDFA), which may be owing to the SNR enhancement ability of an EDFA[37]. Figure 5b summarizes the SCN dependent on ER, showing a clear monotonical decreasing trend. The reduction of crosstalk is as significant as about 28 dB when the ER is increased by 40 dB. Numerical simulations based on the same DAS configuration are also performed (see more detail in Supplementary Note 4) for comparison[38]. The simulated results (Fig. 5c) reveal an ER-dependent crosstalk behavior very similar to the experimental observation except the absolute values, which is associated with noise level set up in the simulation. As a result, the simulated crosstalk suppression is about 27 dB that is very close to the experiment.

Similarly, when the PZT driving is switched off, the PSD along the whole fiber is measured to determine the noise floor, which is usually considered as the sensitivity of a DAS system[39]. The frequency range for statistics is from 500 Hz to 10 kHz. According to the fitting results on the histogram distributions shown in Fig. 5d, the noise floor also decreases with increasing ER. However, the noise reduction becomes less obvious after 50 dB and the total improvement is only about 5.2 dB (Fig. 5e). As a result, the noise floor is much less sensitive to ER compared with the spatial crosstalk. The lowest noise of −71.2 dB rad²/Hz obtained with the EOM in Fig. 5e corresponds to a strain detection

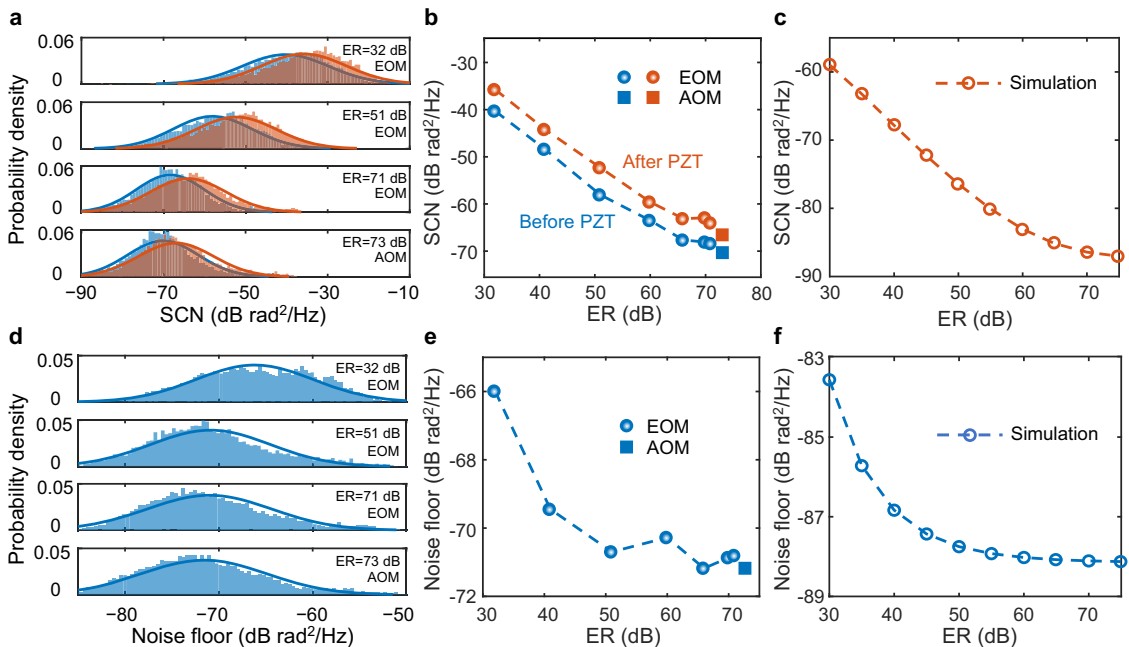

**Fig. 5 | Distributed acoustic sensing (DAS) performances dependent on extinction ratio (ER) of modulators. a** Probability density distribution of measured spatial crosstalk noise (SCN) for different ERs of the electro-optical modulator (EOM) and the acousto-optical modulator (AOM). **b** Average SCN from the fitted distributions against ER. **c** Simulated SCN against ER. **d** Probability density distributions of measured noise floor for different ERs of the EOM and the AOM. **e** Average noise floor from the fitted distributions against ER. **f** Simulated noise floor against ER.

sensitivity of 4 pε/√Hz that is identical to that with the AOM. The achieved sensitivity indicates that the on-chip device is adaptable to a high-performance DAS system. Again, we calculate the noise floor against ER with the same model as before and obtain a trend (Fig. 5f) similar to the experiment result.

## Discussion

The demonstrated coupled-microring on silicon is a representative of coupled-resonator structures whose characteristics including transmission, dispersion, group delay, etc. can be defined flexibly by their structural parameters[40]. Benefitting from the engineered evanescent optical coupling between the rings and input/output waveguides, the bandpass filtering of the coupled-microring shows a large out-of-band rejection (>60 dB) and a steep roll-off (~100 dB/nm). These characteristics are critical to realize the ultra-high ER modulation. Through the carrier injection scheme, high modulation efficiency is achieved at the cost of modulation speed. The 6.5-ns response is associated with the carrier relaxation time and thermo-optic effect in silicon, which are intrinsic to the current EOM design. A number of previous studies on optical switch with silicon PIN structures also give their response times of a few ns[41,42]. The limited modulation speed can be a bottleneck for certain distributed sensing applications where sub-meter spatial resolution[43] or high baud-rate coding pulses[44] are required. However, the short pulses can neither be directly generated by an AOM as the acousto-optical response is usually in the scale of 10 ns. Under the carrier injection scheme, we believe that the EOM could still be faster by proper optimization on doping and thermal management but probably would not go much beyond the limitation of carrier relaxation time in silicon. For example, higher RC bandwidth can be expected by adding intermediate doped sections between the intrinsic silicon waveguide and heavily doped regions to reduce the resistance of the EOM[45].

Regarding the thermal sensitivity of silicon photonic devices, a common approach is to place integrated micro-heaters above onto the rings to compensate temperature-induced drift as well as fabrication errors[46]. The reported heating power is down to a few mW for π phase

shift[47] (see Supplementary Note 5 for more analysis). Recently, thermal trimming with Ge-implanted waveguides on ring modulators also provides a one-off method for wavelength calibration[48,49]. Therefore, the thermal sensitivity issue can be well addressed by existing techniques.

While the demonstrated EOM featuring ultra-high ER, low power consumption and high compactness is highly desired for DAS systems, it may find additional applications in quantum key distribution[50,51], lidar[52], etc., where ER is also crucial for SNR of these systems. A common principle behind the ER requirement is that the leakage light or photons during modulation can contribute to noise when the detection is interference involved. Employment of the on-chip EOM could thus benefit the power efficiencies and footprints of these systems while the SNR would not degrade. Beside the thermal sensitivity, a few more technical limitations of the EOM will need to be addressed before its practical applications in commercial systems. For instance, the nonlinear absorption of silicon waveguides would reduce the EOM transmission significantly when the input optical power goes up to tens of mW[53,54], leading to the requirement of amplification on the modulated output. Additionally, the third-order nonlinearity (Kerr effect) of silicon may induce self-phase modulation[55] and shift the original transmission spectrum. These situations could be prevented by using multimode waveguide instead of single-mode waveguide to reduce the optical intensity that directly determines the strength of the nonlinear effects[56]. Hence, careful adiabatic waveguiding design will be necessary to suppress higher order mode excitation.

Beside SOI, other material platforms such as lithium niobate on insulator and III-V semiconductors (GaAs, InP, etc.) have also shown potential for highly-performing EOMs thanks to multiple excellent properties[57,58]. Targeting high ER modulation alone for DAS application, they will also be suitable and beneficial with proper device designs. On the other hand, if more devices or circuits are expected on one (or as few as possible) material platform for more functionalities, higher throughput but less fabrication complexity, the factors of optical loss, efficiency, footprint, processes etc. have to be considered simultaneously with certain tradeoffs. With its own limitations as

**Table 1 | Performance comparison of typical EOMs on various platforms**

| Structure | Material | ER (dB) | Size[a] | Power dissipation | BW (GHz) | $V_\pi L$ (V·cm) |
|---|---|---|---|---|---|---|
| Racetrack[63] | Plasmonics + EO polymer | 5.2 | R = 10 μm | 12.3–28.8 fJ/bit | 176 | 0.015 |
| MZI[64] | III-V | / | L = 4 mm | <2.5 W | 80 | 0.6 |
| MZI[65] | III-V | / | L = 8 mm | 0.57–0.85 fJ/bit | 54 | 0.55 |
| MZI[66] | SOI + LN | 11.8 | L = 5 mm | 700 fJ/bit | >70 | 2.2 |
| FP cavity[67] | LNOI | <3 | L = 50 μm | 4.5 fJ/bit | >110 | 3.57[b] |
| MZI[68] | LNOI | / | L = 15 mm | 120 fJ/bit | >67 | 2.5 |
| MOS ring[18] | SOI | 5.8 | R = 15 μm | / | >50 | <1.3 |
| MZI[69] | SOI | 3.15 | L = 124 μm | / | 110 | / |
| Single ring[70] | SOI | / | R = 8 μm | / | >60 | 0.825 |
| Coupled rings (this work) | SOI | 68 | R = 10 μm | 3.6 mW | 0.72 | 0.0032[c] |

[a] L indicates the MZI length and R is the ring radius.
[b, c] The modulation efficiency is estimated according to the resonance shift per voltage.

described before, SOI still has good efficiency and transparency as well as large-scale CMOS fabrication capability, which may be more helpful for high-yield and high-density integration. Therefore, in this work, the ultra-high ER modulators are implemented on SOI as a start for on-chip DAS modules/systems. Table 1 summarizes the typical performance metrics of EOMs implemented on different material platforms. It can be seen that high bandwidth is targeted by the vast majority of EOMs. Our work demonstrates the opportunity of EOMs with ultra-high ER for distributed optical fiber sensing applications where bandwidth is less critical.

Due to the frequency shifting feature of AOMs, many DAS systems have employed heterodyne configurations to detect the beat signal from the combination of RBS and local reference[59]. At the same time, homodyne and direct detections are also frequently used to realize high-performance DAS[14,44,60]. An ultra-high ER EOM without frequency shifting will be desired in these cases where otherwise an extra AFS is needed in addition to an AOM. Conventional bulky EOMs are usually based on electro-optical crystals (e.g., lithium niobate[58]). However, complicated configurations such as cascading EOMs and peripheral circuits are necessary to obtain high ER for distributed optical fiber sensing[13]. In contrast, the on-chip EOM reported in this work is much simpler for usage in practice. On the other hand, several reports on DAS systems with frequency multiplexing have demonstrated enhanced SNR[61] and response bandwidth[62]. The frequency multiplexing functionality in the specific systems can be achieved much easier using AOMs than EOMs. Therefore, we believe that the usage of AOMs or EOMs is actually dependent on the system configuration requirement. In those systems whose performances can be boosted by frequency shifting, AOMs are apparently a better option. Otherwise, the factors for consideration will be among ER, power consumption, footprint, etc. and thus the on-chip EOM would be advantageous over an AOM.

In summary, we demonstrate an on-chip silicon EOM with an ultra-high ER up to 68 dB, which is measured from the modulated optical pulse in real time. The power consumption of the EOM for pulse generation is only 3.6 mW, which is two orders of magnitude lower than that of an AOM. The fabrication is fully compatible to standard CMOS foundry process with large-scale manufacturing capability. Furthermore, the packaged EOM is successfully applied in a DAS system, presenting an ultra-high sensitivity of −71.2 dB rad²/Hz (4 pε/√Hz) and a low SCN of −68.1 dB rad²/Hz at 0.5-rad phase change amplitude. The performances are nearly the same as those of a DAS system using a commercial AOM while the compactness and power efficiency are remarkably improved with the on-chip EOM. The SCN and sensitivity dependences on ER are also characterized by experiment rigorously, showing clear evidences for the importance of ER to DAS performances. The ultra-high ER on-chip EOM would be essential for realization of compact and low power DAS systems to be deployed on small

platforms. It also facilitates the development of opto-electronic engines with low-noise lasers, modulators, detectors, etc. to be integrated together on photonic chips for next-generation ultra-compact DAS systems and other distributed optical fiber sensing systems based on OTDR.

## Methods
### On-chip EOM characterization
The on-chip EOMs use grating couplers for optical input/output with a fiber tilting angle of 5°. Straight reference waveguides are fabricated nearby to calibrate the on-chip insertion loss. An RF GSG probe (GGB 40 A) is employed to deliver DC/AC driving signal to the EOM. The modulated output is sent to the power meter or a photodetector (EOT ET-3010) for static or dynamic characterization, respectively.

### DAS setup and operation
CW light from a narrow linewidth laser (NKT Photonics, E15 Erbium Fiber Laser, linewidth <0.1 kHz) is modulated into a pulse train at 20-kHz repetition by the EOM or AOM. The packaged EOM uses edge couplers for wider transmission bandwidth and smaller wavelength sensitivity in optical input/output. Then the pulses are amplified by an EDFA and launched into the sensing fiber. As the transmittance of modulator differs with ERs, a variable optical attenuator (VOA) is used to maintain a same peak power of 200 mW. The amplified spontaneous emission of the amplified RBS is suppressed by a bandpass filter (WL Photonics Inc.) with 0.11-nm bandwidth. Outputs of the three APDs are sampled at 125 MS/s and recorded by the oscilloscope using 20-MHz port bandwidth. As the APD gains are slightly different in practice, the signal levels of the three channels are manually balanced before demodulation processing.

## Data availability
The data that support the findings of this study are available from the corresponding author upon request.

## Code availability
The code that support the simulations of this study are available from the corresponding author upon request.

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

## Acknowledgements

This work was supported by Young Scientists Fund of National Natural Science Foundation of China (62105301, received by B.C.), National Key Research and Development Program of China (2021ZD0109904, received by S.Y.), Key Project of National Natural Science Foundation of China (U21A20453, received by Y.R.), Key Research Project of Zhejiang Laboratory (2020ME0AD02, received by L.M.) and Research Initiation Project of Zhejiang Laboratory (2021ME0PI01, received by B.C.). The authors also thank Dr. Weiwei Zhang in University of Southampton and Dr. Yong Hu in Zhejiang Laboratory for helpful discussions.

## Author contributions

B.C. and Y.R. conceived the idea and supervised the study. B.C. and Z.C. designed the EOMs. B.C., Z.C., and C.S. programed and drawn the device layout. S.Y., C.S., M.W., H.L., and L.L. assisted with the MPW fabrications. B.C. and Z.C. performed the device characterization. X.S. and L.M. built the DAS system. X.S., L.M., and Z.C. performed the DAS experiment and data analysis. C.L. performed the numerical simulation. B.C., Z.C., X.S., L.M., Y.R., and C.L. prepared the manuscript with contributions from all the authors.

## Competing interests

The authors declare no competing interests.

B. C. and Y. R. are the inventors of an awarded invention patent of China (ZL202111352077.1) on the ultra-high ER electro-optical modulator based on coupled ring resonators. The remaining authors declare no competing interests.
