## [Peer Review File · Nature Communications]

Reviewers' Comments:

Reviewer #1:

Remarks to the Author:

In the submitted manuscript, the authors present a groundbreaking development in the field of fiber-optic DAS. They introduce an on-chip silicon EOM as an alternative to traditional AOMs, which are currently used in DAS systems. The EOMs are described as having an ultra-high extinction ratio (ER) of up to 68 dB, a feature crucial for achieving high-performance DAS. These EOMs, based on multiple coupled microrings, are far more compact and energy-efficient than conventional AOMs, and the authors successfully apply them in a DAS system to achieve comparable sensitivity and spatial crosstalk noise to those using AOMs. This work is framed as a major step towards the development of next-generation, ultra-compact, and energy-efficient DAS systems by integrating on-chip opto-electronic devices. In my opinion, the manuscript deserves publication in Nature Communication after mandatory revision by the authors.

I have a few criticisms to be addressed in the revised manuscript.

1/ The paper claims to demonstrate that the EOM can replace AOMs in DAS systems without loss of performance. However, are there any limitations or drawbacks to this? What conditions or applications are not suitable for replacing AOMs with EOMs?

2/ One of the critical questions arising in this concern is why the performance of the DAS system with AOMs is worse (in one of the cases) than that with EOMs, even though EOMs have a higher ER. Is this a limitation of the EOMs or the DAS system itself? Could you provide more insight into this discrepancy?

3/ The paper notes that high modulation efficiency is achieved at the cost of modulation speed (6.5-ns response). Is this limitation intrinsic to the current design? Could this be a bottleneck for certain applications? The manuscript could benefit from discussing these trade-offs more explicitly. It would be beneficial for the authors to discuss any limitations related to modulation speed, particularly in comparison with AOMs.

4/ The discussion section might benefit from an expanded section on the potential future applications of these EOMs in DAS systems and any existing technical limitations that need to be addressed.

5/ The paper could be strengthened by including a more exhaustive comparison of the EOM with existing technologies, such as other types of modulators, in terms of performance metrics other than just ER, size, and power consumption.

6/ While the abstract notes that the DAS is "highly demanded in practice," a section discussing the scientific DAS applications would be beneficial [doi:10.3390/s22031033].

7/Figure 5 Caption - Consider correction: "for difference ERs of the EOM and the AOM"

Reviewer #2:

Remarks to the Author:

In the manuscript "On-chip silicon electro-optical modulator with ultra-high extinction ratio for fiber-optic distributed acoustic sensing" by Z. Cheng et al., the authors study via simulations and experiments a microring-based electro-optic (EO) modulator based on carrier injection. Using a fourth-order microring filter, they are able to achieve a large out-of-band rejection which translates into an exceptionally-high dynamic extinction ratio (ER) of 68 dB. Subsequently, they incorporate the EO modulator within a distributed acoustic sensing (DAS) system and demonstrate performance that matches that of conventional implementations with an acousto-optical modulator (AOM), while achieving significant improvements in footprint and power consumption. The paper is professionally written, the experiments are carefully performed and are supported with simulation results and theoretical discussion. Overall, it is good paper that is expected to interest the

community. I can recommend publication in Nature Communications with the following minor suggestions.

1. In Fig. 1(f) the authors show a modulated optical pulse signal with 200-ns duration. However, in the introduction (p.2, line 71), it is mentioned that durations of 10-100 ns are customary for DAS systems. Why have they not targeted such a pulse duration? The authors are kindly asked to comment.

2. Besides SOI, other material platforms have shown potential for highly-performing electro-optic modulators as well, e.g. Lithium niobate on insulator or III-V semiconductors. Can the authors please comment on whether such platforms have been considered and if they could perhaps be beneficial for the DAS application they are targeting?

3. In a very recent review paper, the state of the art and the underlying physics of integrated electro-optic modulators have been thoroughly documented, see [DOI: 10.1063/5.0048712]. The authors can mention for completeness and check how their performance metrics compare against other modulator implementations.

4. As mentioned by the authors, heaters can be incorporated above the rings for thermal stability. In this case, how would the power consumption be affected? The authors are kindly asked to comment.

Responses to comments on NCOMMS-23-38601

Reviewer #1 (Remarks to the Author):

In the submitted manuscript, the authors present a groundbreaking development in the field of fiber-optic DAS. They introduce an on-chip silicon EOM as an alternative to traditional AOMs, which are currently used in DAS systems. The EOMs are described as having an ultra-high extinction ratio (ER) of up to 68 dB, a feature crucial for achieving high-performance DAS. These EOMs, based on multiple coupled microrings, are far more compact and energy-efficient than conventional AOMs, and the authors successfully apply them in a DAS system to achieve comparable sensitivity and spatial crosstalk noise to those using AOMs. This work is framed as a major step towards the development of next-generation, ultra-compact, and energy-efficient DAS systems by integrating on-chip opto-electronic devices. In my opinion, the manuscript deserves publication in Nature Communication after mandatory revision by the authors.

I have a few criticisms to be addressed in the revised manuscript.

1/ The paper claims to demonstrate that the EOM can replace AOMs in DAS systems without loss of performance. However, are there any limitations or drawbacks to this? What conditions or applications are not suitable for replacing AOMs with EOMs?

Response:

We thank the reviewer for such a professional question. In indeed, one of the key differences between EOM and AOM is that the AOM shifts the frequency of the input light while carving it into pulses. This capability offers additional benefits such as frequency multiplexed interrogation, which can help to reduce the noise level or improve the bandwidth of the sensing system. Researchers from US Naval Research Lab built a DAS system using three AOMs [Sci. Rep. 11, 17921, 2021], which generated a frequency-shifted pulse train that produced uncorrelated Rayleigh backscattered signals. By combining these signals, they were able to reduce the noise floor of the system as well as improve the linearity of the system's response. Frequency shifting can also help to improve the system bandwidth by frequency multiplexing [Opt. Lett. 47, 3, 529-532, 2022]. These benefit in the specific systems can be easily achieved using AOMs. Though it is also possible to use EOMs to build similar systems, generating the frequency-shifted pulse train with same level of stability and spectrum flatness is much less straight forward and cost effective, not to mention that the driving control will be much more complex. Considering that there are still a number of DAS system configurations in which frequency shifting is not necessary, we believe that the usage of AOM or EOM is actually dependent on the system requirement. In those systems whose performances can be boosted by frequency shifting, AOMs are apparently a better option. Otherwise, the factors for consideration will be among ER, power consumption, footprint, etc. and thus the on-chip EOM would be advantageous over an AOM.

The discussion above is summarized and added into the revised discussion section in the main text to further clarify the suitable application scenarios of AOMs and EOMs, respectively.

2/ One of the critical questions arising in this concern is why the performance of the DAS system with AOMs is worse (in one of the cases) than that with EOMs, even though EOMs have a higher

ER. Is this a limitation of the EOMs or the DAS system itself? Could you provide more insight into this discrepancy?

Response:

We thank the reviewer for expressing the concern of discrepancy. We would like to take this opportunity to clarify what we have demonstrated in the manuscript. As indicated by the legends in Figs. 5(a) and 5(d), ER (71 dB) of the EOM is still slightly less than that of the AOM (73 dB) after EDFA in the DAS system. The ER of the commercial AOM is fixed by its specific driver so we only have one value. Therefore, we observe that the spatial crosstalk noise (SCN, ball symbols) with the EOM is also slightly higher than that (revised as solid square symbols) with the AOM, as shown in Fig. 5(b). Both the SCN and noise floor performances of the DAS system with the AOM are not worse than those with the EOM. We humbly guess that the reviewer might had taken the blue and red curves as system SCN with the EOM and the AOM, respectively, by mistake. They actually correspond to the SCN before and after the PZT as the acoustic wave source, respectively. After the PZT, the noise level slightly increases because of non-negligible connection losses between the stretching fiber and the two fiber spools, leading to a separation between the two curves. As a result, limitations of the EOMs or the system are not involved here. To further increase the clarity, we rearrange the legends in Figs. 5(a) and 5(d), and change the diamond symbols into solid squares in Figs. 5(b) and 5(e) to indicate the AOM's results better.

Fig. 5 DAS performances dependent on ER of modulators. **a** Probability density distribution of measured spatial crosstalk noise (SCN) for different ERs of the EOM and the AOM. **b** Average SCN from the fitted distributions against ER. **c** Simulated SCN against ER. **d** Probability density distributions of measured noise floor for different ERs of the EOM and the AOM. **e** Average noise floor from the fitted distributions against ER. **f** Simulated noise floor against ER.

3/ The paper notes that high modulation efficiency is achieved at the cost of modulation speed (6.5-ns response). Is this limitation intrinsic to the current design? Could this be a bottleneck for certain applications? The manuscript could benefit from discussing these trade-offs more explicitly. It would be beneficial for the authors to discuss any limitations related to modulation speed, particularly in comparison with AOMs.

Response:

As we had mentioned in the paragraph of device design, the phase shifters of the EOM are based on “positively-doped, intrinsic and negatively doped” (PIN) structure (Fig. 1b in the main text). When a forward voltage is applied (i.e., P connected to anode while N connected to cathode) as how the EOM operates for modulation, electric current is built up and run through the intrinsic region from P to N (carrier injection), increasing the carrier density in the waveguide substantially. Owing to the carrier plasma dispersion effect (IEEE J. Quantum. Electron. 23, 123-129, 1987), refractive index and optical loss of the waveguide change significantly at once, leading to the observed transmission spectrum shift shown in Fig. 1d of the main text. When the driving signal is removed and no more carrier is injected, the remaining carriers gradually relax and the state of waveguide (refractive index and optical loss) is recovered. On the other hand, the consumed electrical power is mostly transferred to heat and thus affects the refractive index as well via thermo-optic effect. Therefore, the response of the EOM based on PIN structure is highly dependent on the carrier relaxation time and thermal dynamics in the silicon waveguide. As the intrinsic carrier relaxation time is only about 1 ns, we think that the extra time for the rising edge may be induced by the incomplete recovery of thermo-optic effect during the inverted modulation. A number of previous studies on optical switch with silicon PIN structures also give their response times of a few ns (Opt. Express 17, 25, 22271-22280, 2009; Opt. Lett. 47, 11, 2758-2761, 2022). Hence, the 6.5-ns response is basically intrinsic to the current design. Under the carrier injection scheme, we believe that the EOM could still be faster by proper optimization on doping and thermal management but probably would not go much beyond the limitation of carrier relaxation time in silicon. According to the comparison among different kinds of EOMs (in response to Comment #5), we can see that the high-speed modulators on SOI (tens of Gbps) are much less efficient (larger V_{π} -L) than our PIN one. As a result, the speed-efficiency tradeoff is challenging for silicon modulators.

The limited modulation speed of this EOM does have an impact for certain distributed sensing applications. According to principle of optical time domain reflectometry, spatial resolution of the distributed sensing is proportional to duration of optical probe pulses, i.e., $r=ct/2n$, assuming there is no constraint from sensing fiber or interrogator system. Here c is light speed in vacuum, t is pulse duration and n is effective refractive index of the sensing fiber, respectively. Therefore, the modulation speed determines how short the probe pulses can be and thus the spatial resolution under the scheme of external modulation. For example, 1-m resolution can be obtained with 10-ns pulse duration, which can meet most of practical requirements of DAS with our current on-chip EOM. For finer resolution (0.1 m), sub-ns pulses would be necessary. However, such short pulses can neither be generated by an AOM as the acousto-optical response is usually in the scale of 10 ns. Researchers from University of Southampton used 500-ps pulse to demonstrate a DAS system with 0.1-m spatial resolution [Opt. Contin. 1, 9, 2002-2010, 2022], which is an order of magnitude smaller than most of the DAS systems. In that case, high-speed and high-ER modulation is implemented by cascading a semiconductor optical amplifier (SOA) after an EOM. Note that SOA can compromise the coherency of low-noise laser source which is critical for most DAS systems. Additionally, if a DAS system utilizes high-speed pulse coding for resolution or SNR enhancement (Opt. Express 24, 19, 22303-22318, 2016), the modulator bandwidth will have to exceed 1 GHz.

Through the discussion above, we can see that the required modulation speed is highly associated with DAS system specifications. While the speed of our on-chip EOM already satisfies

most DAS applications, higher performance in term of resolution (cm scale) needs a faster modulator that is usually a bulky lithium niobate modulator with significant size and power consumption. The choice for DAS systems is thus dependent on the specific performance being focused (e.g., resolution, size, efficiency etc.). The whole discussion here is summarized properly and incorporated into the revised discussion section in the main text.

4/ The discussion section might benefit from an expanded section on the potential future applications of these EOMs in DAS systems and any existing technical limitations that need to be addressed.

Response:

We appreciate this constructive suggestion from the reviewer and therefore add a discussion about the potential future applications of these EOMs in DAS systems and any existing technical limitations as below, which makes a single paragraph in the revised discussion section.

While the demonstrated EOM featuring ultra-high ER, low power consumption and high compactness is highly desired for DAS systems, it may find additional applications in quantum key distribution (Opt. Express 23, 13, 17511-17519, 2015; Phys. Rev. Appl. 9, 054008, 2018), lidar (Appl. Opt. 60, 6, 1623-1628, 2021; J. Light. Technol. 39, 14, 4661-4670, 2021), etc., where ER is also crucial for SNR of these systems. A common principle behind the ER requirement is that the leakage light or photons during modulation can contribute to noise when the detection is interference involved. Employment of the on-chip EOM could thus benefit the power efficiencies and footprints of these systems while the SNR would not degrade. Beside the thermal sensitivity, a few more technical limitations of the EOM will need to be addressed before its practical applications in commercial systems. For example, the nonlinear absorption of silicon waveguides would reduce the EOM transmission significantly when the input optical power goes up to tens of mW (Opt. Express 12, 8, 1611-1621, 2004; Opt. Lett. 31, 11, 1714-1716, 2006), leading to the requirement of amplification on the modulated output. Additionally, the third-order nonlinearity (Kerr effect) of silicon may induce self-phase modulation (Appl. Phys. Lett. 80, 3, 416-418, 2002) and shift the original transmission spectrum. These situations could be prevented by using multimode waveguide instead of single-mode waveguide to reduce the optical intensity that directly determines the strength of the nonlinear effects (Opt. Lett. 48, 14, 3729-3732, 2023). Hence, careful adiabatic waveguiding design will be necessary to suppress higher order mode excitation.

5/ The paper could be strengthened by including a more exhaustive comparison of the EOM with existing technologies, such as other types of modulators, in terms of performance metrics other than just ER, size, and power consumption.

Response:

This suggestion from the reviewer is very helpful. Therefore, we make the comparison as summarized by the table below, to present the main performance metric difference of integrated EOMs implemented with various technologies. This table is also added into the revised discussion section in the main text.

Structure	Material	ER (dB)	Size ^(a)	Power dissipation	BW (GHz)	$V_{\pi}L$ (V • cm)
Racetrack [1]	Plasmonics + EO polymer	5.2	R=10 μm	12.3-28.8 fJ/bit	176	0.015
MZI [2]	III-V	/	L=4 mm	<2.5 W	80	0.6
MZI [3]	III-V	/	L=8 mm	0.57-0.85 fJ/bit	54	0.55
MZI [4]	SOI + LN	11.8	L=5 mm	700 fJ/bit	>70	2.2
FP cavity [5]	LNOI	<3	L= 50 μm	4.5 fJ/bit	>110	3.57 ^(b)
MZI [6]	LNOI	/	L=15 mm	120 fJ/bit	>67	2.5
MOS ring [7]	SOI	5.8	R=15 μm	/	>50	<1.3
MZI [8]	SOI	3.15	L=124 μm	/	110	/
Single ring [9]	SOI	/	R=8 μm	/	>60	0.825
Coupled rings (this work)	SOI	68	R=10 μm	3.6 mW	0.72	0.0032 ^(c)

(a) L indicates the MZI length and R is the ring radius.

(b, c) The modulation efficiency is estimated according to the resonance shift per voltage.

References

- [1] Eppenberger, Marco, et al. "Resonant plasmonic micro-racetrack modulators with high bandwidth and high temperature tolerance." *Nature Photonics* 17.4 (2023): 360-367.
- [2] Ogiso, Yoshihiro, et al. "80-GHz bandwidth and 1.5-V V_{π} InP-based IQ modulator." *Journal of Lightwave Technology* 38.2 (2019): 249-255.
- [3] Lange, Sophie, et al. "100 GBd intensity modulation and direct detection with an InP-based monolithic DFB laser Mach–Zehnder modulator." *Journal of Lightwave Technology* 36.1 (2018): 97-102.
- [4] He, Mingbo, et al. "High-performance hybrid silicon and lithium niobate Mach–Zehnder modulators for 100 Gbit s⁻¹ and beyond." *Nature Photonics* 13.5 (2019): 359-364.
- [5] Pan, Bing-Cheng, et al. "Ultra-compact lithium niobate microcavity electro-optic modulator beyond 110 GHz." *Chip* 1.4 (2022): 100029.
- [6] Xu, Mengyue, et al. "High-performance coherent optical modulators based on thin-film lithium niobate platform." *Nature communications* 11.1 (2020): 3911.
- [7] Zhang, Weiwei, et al. "Harnessing plasma absorption in silicon MOS ring modulators." *Nature Photonics* 17.3 (2023): 273-279.
- [8] Han, C., et al. "Ultra-Compact Silicon Modulator with 110 GHz Bandwidth. 2022 Optical Fiber Communications Conf. and Exhibition (OFC). Diego, USA, 06–10 March 2022." (2022).
- [9] Zhang, Yuguang, et al. "240 Gb/s optical transmission based on an ultrafast silicon microring modulator." *Photonics Research* 10.4 (2022): 1127-1133.

6/ While the abstract notes that the DAS is "highly demanded in practice," a section discussing the scientific DAS applications would be beneficial [doi:10.3390/s22031033].

Response:

We agree with the reviewer about discussing the scientific applications of DAS to emphasize the importance of developing high-performance DAS systems. As the current work mainly focuses on the ultra-high ER on-chip EOM and its application for DAS, we would prefer constraining this

discussion at the beginning of the introduction, which is also given below.

Besides, the DAS technology also provides a novel approach for a number of natural scientific research directions (Sensors 22, 1033, 2022) ranging from insect monitoring (Sensors 21, 1592, 2021), ice activity study in Arctic sea (Geophys. Res. Lett. 49, 24, e2022GL099880, 2022) to optical microphone (Opt. Express 24, 26, 29597-29602, 2016), etc. Such scientific DAS applications indicate the exceptional capability of observing the nature via acoustic waves.

7/Figure 5 Caption - Consider correction: "for difference ERs of the EOM and the AOM".

Response:

We thank the reviewer for pointing out the pen slip. The "difference" in the caption ought to be "different", which are corrected in the revised version. We are sorry for the carelessness.

Reviewer #2 (Remarks to the Author):

In the manuscript "On-chip silicon electro-optical modulator with ultra-high extinction ratio for fiber-optic distributed acoustic sensing" by Z. Cheng et al., the authors study via simulations and experiments a microring-based electro-optic (EO) modulator based on carrier injection. Using a fourth-order microring filter, they are able to achieve a large out-of-band rejection which translates into an exceptionally-high dynamic extinction ratio (ER) of 68 dB. Subsequently, they incorporate the EO modulator within a distributed acoustic sensing (DAS) system and demonstrate performance that matches that of conventional implementations with an acousto-optical modulator (AOM), while achieving significant improvements in footprint and power consumption. The paper is professionally written, the experiments are carefully performed and are supported with simulation results and theoretical discussion. Overall, it is good paper that is expected to interest the community. I can recommend publication in Nature Communications with the following minor suggestions.

1. In Fig. 1(f) the authors show a modulated optical pulse signal with 200-ns duration. However, in the introduction (p.2, line 71), it is mentioned that durations of 10-100 ns are customary for DAS systems. Why have they not targeted such a pulse duration? The authors are kindly asked to comment.

Response:

We thank the reviewer for this comment. The observed pulse in Fig. 1(f) with 200-ns duration is mainly for the demonstration of pulse modulation functionality using the fabricated EOM. Particularly, because of the limited resolution bandwidth (10 MHz) of our ESA, the self-heterodyne measurement for ultra-high ER characterization requires over-200-ns pulse duration to fully resolve the pulse profile along the vertical direction. When the EOM is applied to our DAS system, the duration is reduced to 100 ns as stated in the first paragraph of the Device application section. Actually, with the current response, we can obtain shorter pulses easily by adjusting the driving duty cycle, as given by Fig. R1 below. A small spike is observed next to the falling edge, which is owing to a little instability of the driving signal with sharp edges. To avoid any possible confusion, this 20-ns waveform is also added into Fig. 1(f) to indicate the ability of pulse duration adjustment.

Fig. R1. Modulated optical pulse signal with 20-ns duration from the EOM.

2. Besides SOI, other material platforms have shown potential for highly-performing electro-optic modulators as well, e.g. Lithium niobate on insulator or III-V semiconductors. Can the authors please comment on whether such platforms have been considered and if they could perhaps be beneficial for the DAS application they are targeting?

Response:

We agree with the reviewer on the selection of more material platforms for highly-performing electro-optic modulators (EOMs). Today, most commercial transceiver modules for optical communications are still using III-V semiconductors (GaAs, InP, etc.) thanks to the high EO efficiency for modulation and low dark current for photodetection. Recently, thin film lithium niobate on insulator (LNOI) also demonstrated significant advantages on EO bandwidth, linear phase modulation and ultralow loss. Targeting high ER modulation alone for DAS application, both types of materials will also be suitable and beneficial with proper device designs. We did consider developing the EOM based on LNOI and are actually working on it now. III-V semiconductors is beyond the reach of our capability and hence not being considered currently. On the other hand, if more devices or circuits are expected to be integrated into one (or as few as possible) material platform for more functionalities but less fabrication complexity, the factors of optical loss, efficiency, footprint etc. have to be considered simultaneously with certain tradeoffs. SOI does have its own limitations as described for example in the response to Comment #4 from Reviewer #1. Meanwhile, it shows better capability of large-scale CMOS fabrication that may be more helpful for high-yield and high-density integration. Therefore, we considered SOI as a promising candidate. The choice of material is of course not definite and constant. We are expecting more kinds of on-chip optoelectronic devices for distributed optical fiber sensing from the community, which will boost the research and development for sure.

This discussion is summarized and added as a single paragraph into the revised discussion section in the main text.

3. In a very recent review paper, the state of the art and the underlying physics of integrated electro-optic modulators have been thoroughly documented, see [DOI: 10.1063/5.0048712]. The authors can mention for completeness and check how their performance metrics compare against other modulator implementations.

Response:

This suggestion is much appreciated. The recommended reference is cited in the introduction section and we summarize the performance metrics of our EOM in comparison with multiple typical works on modulators mentioned by this review paper. The table for comparison is given below, which is also added into the revised discussion section in the main text.

Structure	Material	ER (dB)	Size ^(a)	Power dissipation	BW (GHz)	$V_{\pi}L$ (V • cm)
Racetrack [1]	Plasmonics + EO polymer	5.2	R=10 μm	12.3-28.8 fJ/bit	176	0.015
MZI [2]	III-V	/	L=4 mm	<2.5 W	80	0.6
MZI [3]	III-V	/	L=8 mm	0.57-0.85 fJ/bit	54	0.55
MZI [4]	SOI + LN	11.8	L=5 mm	700 fJ/bit	>70	2.2
FP cavity [5]	LNOI	<3	L= 50 μm	4.5 fJ/bit	>110	3.57 ^(b)
MZI [6]	LNOI	/	L=15 mm	120 fJ/bit	>67	2.5
MOS ring [7]	SOI	5.8	R=15 μm	/	>50	<1.3
MZI [8]	SOI	3.15	L=124 μm	/	110	/
Single ring [9]	SOI	/	R=8 μm	/	>60	0.825
Coupled rings (this work)	SOI	68	R=10 μm	3.6 mW	0.72	0.0032 ^(c)

(a) L indicates the MZI length and R is the ring radius.

(b, c) The modulation efficiency is estimated according to the resonance shift per voltage.

References

- [1] Eppenberger, Marco, et al. "Resonant plasmonic micro-racetrack modulators with high bandwidth and high temperature tolerance." *Nature Photonics* 17.4 (2023): 360-367.
- [2] Ogiso, Yoshihiro, et al. "80-GHz bandwidth and 1.5-V V_{π} InP-based IQ modulator." *Journal of Lightwave Technology* 38.2 (2019): 249-255.
- [3] Lange, Sophie, et al. "100 GBd intensity modulation and direct detection with an InP-based monolithic DFB laser Mach–Zehnder modulator." *Journal of Lightwave Technology* 36.1 (2018): 97-102.
- [4] He, Mingbo, et al. "High-performance hybrid silicon and lithium niobate Mach–Zehnder modulators for 100 Gbit s⁻¹ and beyond." *Nature Photonics* 13.5 (2019): 359-364.
- [5] Pan, Bing-Cheng, et al. "Ultra-compact lithium niobate microcavity electro-optic modulator beyond 110 GHz." *Chip* 1.4 (2022): 100029.
- [6] Xu, Mengyue, et al. "High-performance coherent optical modulators based on thin-film lithium niobate platform." *Nature communications* 11.1 (2020): 3911.
- [7] Zhang, Weiwei, et al. "Harnessing plasma absorption in silicon MOS ring modulators." *Nature Photonics* 17.3 (2023): 273-279.
- [8] Han, C., et al. "Ultra-Compact Silicon Modulator with 110 GHz Bandwidth. 2022 Optical Fiber Communications Conf. and Exhibition (OFC). Diego, USA, 06–10 March 2022." (2022).
- [9] Zhang, Yuguang, et al. "240 Gb/s optical transmission based on an ultrafast silicon microring modulator." *Photonics Research* 10.4 (2022): 1127-1133.

4. As mentioned by the authors, heaters can be incorporated above the rings for thermal stability. In this case, how would the power consumption be affected? The authors are kindly asked to comment.

Response:

We thank the reviewer for the valuable comment. Silicon has a relatively higher thermo-optic coefficient ($dn/dT=1.86 \times 10^{-4}/\text{K}$ @300K) than many other integrated photonic materials (e.g., SiN

and LNOI). Hence, thermal tuning for silicon photonic circuit is usually more efficient. A previous study (Opt. Express 18, 19, 20298-20304, 2010) had shown an ultralow tuning power of 1.2 mW for π phase shift (2.4 mW per FSR) in silicon racetrack resonators with air trenches. We also simulate the thermal heating of a 100- μm -long TiN heater on a silicon rib waveguide without air trenches. Fig. R2(a) shows the simulated temperature distribution in the cross-section with 50-mW heating power applied to the TiN strip. With the knowledge of thermo-optic coefficient of silicon (Appl. Phys. Letts 74, 22, 3338, 1999), the induced phase shift of the rib waveguide against heating power is obtained as given in Fig. R2(b). As a result, the power for π phase shift in this case is as large as 16 mW owing to the thermal conduction with the surrounding solid. Therefore, we estimate that the power consumption would increase by 5 mW in maximum if heaters are incorporated above the four rings with air trenches around. In practice, the increment would be less as the ring resonance drift is usually much smaller than half of an FSR. The relatively short perimeter restricts the accumulated phase errors along the ring induced by fabrication errors. Summing up the heating power, the total power dissipation of the coupled-ring EOM is still less than 10 mW, in a substantial contrast to that (Watt-scale) of an AOM.

This discussion with the simulation results are added into the supplementary information.

Fig. R2 (a) Simulated temperature distribution in the cross-section with 50-mW heating power applied to the TiN strip. (b) Simulated phase shift of the rib waveguide dependent on heating power.

Reviewers' Comments:

Reviewer #1:

Remarks to the Author:

The authors' revision has improved the quality of the manuscript. In my opinion it deserves the publication in Nature Communications as it is.

Reviewer #2:

Remarks to the Author:

The authors have thoroughly addressed all the reviewers concerns and have revised the manuscript accordingly. I can recommend publication to Nature Communications. One final point concerning comment #3 of Reviewer #2. While the authors make use of the suggested review paper (J. Appl. Phys. 130, 010901 (2021), DOI: 10.1063/5.0048712), it does not appear in the reference list of the revised paper.

Response to Referees

Reviewer #1 (Remarks to the Author):

The authors' revision has improved the quality of the manuscript. In my opinion it deserves the publication in Nature Communications as it is.

Response

We sincerely thank the reviewer for the approval to the revised manuscript.

Reviewer #2 (Remarks to the Author):

The authors have thoroughly addressed all the reviewers concerns and have revised the manuscript accordingly. I can recommend publication to Nature Communications. One final point concerning comment #3 of Reviewer #2. While the authors make use of the suggested review paper (J. Appl. Phys. 130, 010901 (2021), DOI: 10.1063/5.0048712), it does not appear in the reference list of the revised paper.

Response

We sincerely thank the reviewer for the approval to the revised manuscript. Meanwhile, we are sorry about the missing recommended reference in the reference list, which has been added as Ref. 20 and highlighted in yellow in the revised main text.